# Long-Term Follow-Up of Inpatients with Failed Back Surgery Syndrome Who Received Integrative Korean Medicine Treatment: A Retrospective Analysis and Questionnaire Survey Study

**DOI:** 10.3390/jcm10081703

**Published:** 2021-04-15

**Authors:** Ju-Hun Park, Kang-Eah Choi, Sang-Gyun Kim, Hui-Yeong Chu, Sang-Woon Lee, Tae-Ju Kim, Hyun-Woo Cho, Sang Don Kim, Kyoung Sun Park, Yoon Jae Lee, Jin Ho Lee, In-Hyuk Ha

**Affiliations:** 1Haeundae Jaseng Hospital of Korean Medicine, Busan 48102, Korea; lovejx0308@nate.com (J.-H.P.); lizzy711@naver.com (K.-E.C.); nobless0511@hanmail.net (S.-G.K.); nancy016@hanmail.net (H.-Y.C.); lsw7563@naver.com (S.-W.L.); bkbygo@naver.com (T.-J.K.); kamui0328@gmail.com (H.-W.C.); donmuta@naver.com (S.D.K.); 2Jaseng Hospital of Korean Medicine, Seoul 06110, Korea; lovepks0116@gmail.com (K.S.P.); jasengjsr@gmail.com (J.H.L.); 3Jaseng Spine and Joint Research Institute, Jaseng Medical Foundation, Seoul 06110, Korea; goodsmile8119@gmail.com

**Keywords:** failed back surgery syndrome, integrative medicine, shinbaro, pharmacopuncture, acupuncture, musculoskeletal manipulations, surveys and questionnaires

## Abstract

Introduction: this study aimed to investigate the long-term clinical efficacy and satisfaction degree of integrative Korean medicine (KM) treatment for patients with failed back surgery syndrome (FBSS). Methods: we performed a follow-up questionnaire survey and retrospective analysis of medical records for patients with FBSS who underwent inpatient treatment for ≥ 1 week. The primary evaluation indices were numeric rating scale (NRS) scores for low back pain (LBP) and leg pain at admission and discharge. Sub-evaluation indices included the Oswestry Disability Index (ODI) and EuroQol 5-dimension (EQ-5D) score. The follow-up questionnaire survey obtained information regarding previous surgeries; reasons for satisfaction/dissatisfaction with surgical and KM treatment; and current status. Results: compared with at admission, there was a significant post-treatment decrease in the NRS scores for LBP and leg pain, as well as the ODI score. Further, there was a significant post-treatment increase in the EQ-5D score. Regarding the patients’ global impression of change for KM treatment administered during admission and at the follow-up questionnaire survey, 101 (95.3%) patients selected “minimally improved” or better. Conclusion: integrative KM treatment could effectively reduce pain, as well as improve function and health-related quality of life, in patients with FBSS.

## 1. Introduction

Approximately 15–20% of all adults experience low back pain (LBP) annually; moreover, 50–80% of adults experience at least one LBP episode over their lifetime [1]. Given the increase in the number of patients with LBP seeking treatment, there has been a significant increase in the number of individuals undergoing surgical treatment in the past two decades [2]. Since the LBP prevalence increases with age, there is a natural increase in the percentage of surgical treatments for LBP with population ageing [3]. Surgical treatments have increased significantly between 1999 and 2013, from 24.5 to 48.83 per 100,000 (*p* < 0.001). The increase was most marked in the oldest age groups with a 2.8-fold increase in procedures for those aged ≥ 60 years in England [4].

Although there have been advances in surgical techniques and equipment, some patients still present with postoperative persistent pain or discomfort, with the prevalence of failed spinal surgery, generally ranging from 10% to 40% [5]. Accordingly, failed back surgery syndrome (FBSS) is considered as “surgical end stage after one or several interventions on the lumbar neuroaxis indicated to relieve lower back pain, radicular pain or the combination of both, without effect” [6]. The International Association for the Study of Pain defines FBSS as “lumbar pain of unknown origin either persisting despite surgical intervention or appearing after surgical intervention for spinal pain originally in the same topographical distribution.” [7].

Patients with FBSS present with a high pain level and low health-related quality of life (HRQoL) resulting from the failure to control existing pain. Patients presenting secondary neuropathic pain due to FBSS experience a greater pain level, a decrease in HRQoL, and functional impairment compared with patients presenting other chronic pain disorders, including complex regional pain syndrome (CRPS), rheumatoid arthritis, osteoarthritis, and fibromyalgia [8]. Moreover, patients with FBSS have been reported to have a work impairment rate of 78% [8], which is significantly higher than that of patients with CRPS (31%) [9] and rheumatoid arthritis (50%) [10]. As indicated by the high work impairment rate and annual medical expenditure, the FBSS-related public burden could grow even further [11].

A multi-disciplinary approach is considered as the most effective treatment for patients with FBSS; further, there is a need for involvement of physicians, psychologists, physical therapists, and other affiliated healthcare professionals to improve the treatment outcomes of patients with FBSS [2]. However, it is difficult to perform treatment using only conventional medical management [12] with complementary and alternative medicine emerging as a treatment option for numerous patients. There have been studies on Korean medicine (KM) treatment for FBSS, including a retrospective review of KM treatment among 707 patients [13] and a prospective observational study on 120 patients with FBSS. The latter study observed that integrative KM treatment allowed favorable long-term outcomes in terms of pain, function, and HRQoL in patients with chronic FBSS lacking a good response to conventional medical management [14]. However, there have been scarce studies on integrative KM treatment for patients with FBSS; moreover, there are no supporting studies required before performing randomized controlled trials (RCTs). This study aimed to investigate long-term clinical efficacy and satisfaction degree of integrative KM treatment, which is a form of conservative treatment, in patients with FBSS.

## 2. Materials and Methods

This multi-center, long-term follow-up observational study investigated patients with FBSS who underwent at least one spinal surgery for symptom improvement followed by inpatient treatment for persistent pain or post-alleviation pain recurrence. We retrospectively reviewed the medical records of patients with FBSS who visited one of four regional network KM hospitals (Gangnam, Daejeon, Bucheon, and Haeundae) between January 2015 and December 2019; moreover, a follow-up questionnaire survey was conducted. The KM hospitals that served as study centers are spine speciality hospitals, as recognized by the Minister of Health and Welfare. Although the primary treatment modality in these hospitals is KM treatment, they pursue an integrative treatment model involving conventional medicine and Oriental medicine based on modern medical diagnostic technology [14,15]. The history of spine surgery was verified by a radiologist and KM doctor based on computed tomography (CT) and lumbar magnetic resonance imaging (MRI) scans obtained at our hospital.

### 2.1. Study Population

#### 2.1.1. Inclusion Criteria

(1) Patients with a history of low back surgery admitted with a chief complaint of persistent pain/discomfort or recurrent pain/discomfort. (2) Patients who underwent inpatient treatment at our hospital for ≥1 week. (3) Patients aged 19–70 years. (4) Patients who voluntarily provided informed consent for study participation.

#### 2.1.2. Exclusion Criteria

(1) Patients diagnosed with severe specific disorders that could cause LBP and leg pain (metastatic spinal tumor, spinal infection, ankylosing spondylitis, and spinal dislocation). (2) Patients with progressive neurological deficits or severe concurrent neurologic symptoms. (3) Patients with non-spinal causes and/or soft tissue problems (tumors, fibromyalgia, rheumatoid arthritis, or gout). (4) Patients with other chronic diseases that could affect the interpretation of the treatment effect/outcome (cardiovascular disease, renal disease, diabetic neuropathy, dementia, or epilepsy). (5) Patients taking corticosteroids, immunosuppressant drugs, psychiatric medicines, or other drugs considered as inappropriate during the inpatient treatment period. (6) Patients previously admitted to our hospital within the past six months. (7) Patients unfit for study participation due to other reasons as determined by the researcher (patient who have a difficulty in communicating with researchers due to physical disability or mental retardation). (8) Patients who did not consent to study participation.

### 2.2. Analysis of Medical Records

#### General Information

Demographic characteristics of interest included age and sex. We collected the following medical information from the electronic medical records (EMRs):
Date and mode of onset (onset dates of current and initial symptoms, as well as reasons for symptom onset).Intervention(s) received between onset and admission (analgesic, steroid, etc.).Current disease history and information regarding spinal surgery (diagnosis of spinal disorder, spinal surgery, name of surgical procedure, and surgical site).Comorbidities (hypertension, diabetes, heart disease, liver disease, and other comorbidities).Social history (smoking, drinking, and occupation).Radiological findings (L-spine T2-weighted MRI or CT scans).

### 2.3. Treatment

Details regarding prescriptions in the EMR were extracted to investigate the number of patients who received different KM treatment types prescribed by a KM doctor, as well as the average number of each treatment. For herbal treatment, one dose was counted as one treatment; however, for all other treatments, one treatment was counted each time it was performed/administered.

### 2.4. Follow-Up Questionnaire Survey

A follow-up questionnaire survey was conducted to obtain information regarding low back surgery and KM treatment; reason(s) for satisfaction/dissatisfaction with each treatment; and current status. This questionnaire survey was performed via telephone interviews and online Google questionnaires. Questionnaire items included surgery history before inpatient KM treatment; postoperative pain recurrence; improvement level from and satisfaction degree with KM treatment; surgery history after inpatient KM treatment; current treatment; and evaluation of current symptoms (using numeric rating scale (NRS), Oswestry Disability Index (ODI), and EuroQol 5-dimension (EQ-5D-5L)). In the satisfaction survey, the subjective satisfaction degree of patients was scored on a 1–10 point scale. Additionally, reasons for satisfaction/dissatisfaction with each treatment were investigated based on multiple-choice items. The choices included the extent and speed of post-treatment pain reduction; functional recovery; treatment ease; recurrence frequency; sequelae; cost burden; and availability of treatment information.

### 2.5. Outcomes

#### 2.5.1. Primary Outcome

Numeric Rating Scale (NRS) [16]

The NRS subjectively measures the pain level on a 0–10 scale, with 0 representing no pain and 10 representing the worst pain imaginable. The visual analogue scale is also widely used; however, the NRS is relatively easier to use since it does not require good vision or motor function. NRS scores for LBP and leg pain at admission and discharge were collected from the EMRs. In the follow-up questionnaire survey, NRS scores were collected at five different time points: onset, after surgery, before admission to the KM hospital, after discharge from the KM hospital, and at present.

#### 2.5.2. Secondary Outcomes

Oswestry Disability Index (ODI) [17]

The ODI is a 10-item questionnaire for assessing the LBP-induced disability degree. Each item is divided into six stages with 0–5 points being assigned to each item. The score is positively correlated with the severity of disability. We used a previously validated Korean version of the ODI questionnaire [18]. ODI scores at admission and discharge were collected from the EMR. In the follow-up questionnaire survey, only the current ODI score was collected.

5-Level EuroQol 5-Dimension (EQ-5D-5L) [19]

The EQ-5D-5L was developed to assess the HRQoL and is widely used across healthcare fields. It is comprised of five dimensions regarding current health status (mobility, self-care, usual activities, pain/discomfort, and anxiety/depression). Moreover, each dimension is assessed through five levels (1, no problems; 2, slight problems; 3, moderate problems; 4, severe problems; and 5, extreme problems). The weight for HRQoL was calculated by applying an estimated weight model for Koreans [20]. EQ-5D-5L scores at admission and discharge were collected from the EMR. Since the EQ-5D-3L was used on patients admitted before November 2017, we applied a weight appropriate for 3L to calculate the scores [21]. In the follow-up questionnaire survey, the current EQ-5D-5L score was collected.

### 2.6. Statement of Ethics

This study was approved by the Institutional Review Board (IRB) of Jaseng Hospital of Korean Medicine (IRB approval No. JASENG 2020-09-017, http://www.e-irb.com/index.jsp, accessed on 29 September 2020). Based on our study objectives, the EMRs were analyzed to obtain data from patients who had provided consent at our hospital. Since our questionnaire survey targeted patients who had been discharged, as well as covered an extensive survey area, it was conducted via telephone interviews or online Google questionnaires. The patients were informed regarding the source of contact information and use of personal information. For patients who provided consent, the survey was conducted; otherwise, the use of information was stopped and all personal information was destroyed.

### 2.7. Statistical Analysis

Continuous variables were presented as mean ± standard deviation (SD) while categorical variables were presented as frequency and percentage. Subsequently, differences in baseline characteristics between patients who completed the follow-up questionnaire and those who did not were assessed using the independent t-test and chi-squared test.

The NRS, ODI, and EQ-5D scores at admission, discharge, and follow-up were presented as mean ± SD. The decrease over time was tested using a linear mixed model. Each time point in the model was considered as a categorical variable and included as a fixed effect with the baseline value of each outcome. The subject was included as a random effect. The results were presented with a 95% confidence interval (CI).

Regarding the minimal clinically important difference (MCID) for patients with FBSS, an NRS score of 2.25 for LBP [22], NRS score of 2.75 for leg pain [22], ODI score of 9 [22], and EQ-5D score of 0.17 [23] were used based on previous studies. Moreover, a ≥ 50% reduction in the baseline pain, which has been used in numerous previous studies, was used as an additional MCID [24,25,26]. Regarding the NRS scores for LBP, there were two MCID criteria. To allow convenience, a ≥50% reduction in the NRS score was set as MCID 1 while a decrease in the NRS score by 2.25 was set as MCID 2. Patients who achieved an MCID were reported as the number (*n*) and percentage (%). Subsequently, we used a multivariate logistic regression model to analyze whether MCID was achieved. The model included the following variables: age, sex, onset, onset mode, surgery type, surgery on L4/5 disc level, disc herniation, stenosis, spondylolisthesis, hypertension, diabetes, smoking, alcohol, NRS scores for arms and legs, and ODI and EQ-5D scores at admission. The results were presented as odds ratio (OR) and 95% CI. The goodness of fit was given as area under the curve (AUC).

Statistical significance was set at a *p* value < 0.05. Statistical analyses were performed using R Studio (Version 1.1.463-© 2009–2018 RStudio, Inc., Boston, MA, USA).

## 3. Results

### 3.1. Study Flow

We included 254 patients who experienced recurrent LBP or leg pain after back surgery with subsequent admission to one of four regional network KM hospitals in this study. Among them, we excluded 20 patients with missing data regarding pain at admission or not meeting the enrolment criteria. Consequently, 234 patients were included in the analysis set. For MCID analysis, 231 patients were included after excluding two and one patient(s) due to missing data regarding the reason for onset and drinking status at admission. Further, 106 patients completed the follow-up questionnaire survey via telephone or internet (Figure 1).

### 3.2. Baseline Characteristics

The mean age of the patients was 54.9 ± 11.5 years; further, there were 141 (60.3%) female patients. The mean duration of hospital stay was 28.1 ± 17.1 days. Regarding the surgery type, laminectomy was most common (*n* = 219, 93.6%) followed by discectomy (*n* = 71, 30.3%). Regarding the operated disc level, L4–5 was most common (*n* = 157, 67.1%). Regarding the lumbar MRI or CT findings, herniation of the nucleus pulposus (HNP) was most common (*n* = 129, 55.1%), followed by spinal stenosis (*n* = 67, 28.6%) (Table 1). There were no significant differences in the demographic characteristics between the response and non-response groups (Appendix A).

### 3.3. Post-Treatment Changes in the Values

There was a significant post-treatment improvement in the NRS, ODI, and EQ-5D scores of patients with FBSS. The NRS score for LBP decreased by 2.62 points (from 5.77 ± 1.38 points at admission to 3.15 ± 1.49 points at discharge; 95% CI: 2.41–2.82). The NRS score for leg pain decreased by 1.88 points (from 4.40 ± 2.76 points at admission to 2.52 ± 1.99 points at discharge; 95% CI: 1.68–2.08). These findings were indicative of a significant improvement in pain (Figure 2A) (*p* < 0.001). In the follow-up questionnaire survey, the mean NRS score at onset, after surgery, before admission to KM hospital, after discharge from KM hospital, and present time were 8.54 ± 1.45, 4.35 ± 2.75, 7.00 ± 1.67, 3.53 ± 2.03, and 3.54 ± 2.49, respectively (Figure 2B).

Moreover, we observed a significant functional improvement indicated by a decrease in the ODI score by 17.35 points (at admission (50.55 ± 16.53), at discharge (33.19 ± 15.93); 95% CI: 15.09–19.62) and 23.50 points (at admission (50.55 ± 16.53), at questionnaire survey (27.39 ± 16.97); *p* < 0.001). Further, there was a significant improvement in the HRQoL indicated by an increase in the EQ-5D score by 0.20 points (at admission (0.54 ± 0.22), at discharge (0.74 ± 0.13); 95% CI: 0.17–0.24; *p* < 0.001; Table 2).

### 3.4. MCID Analysis

The number of patients who had achieved MCID 1 and 2 in the NRS score for LBP at discharge was 96 (41.6%) and 109 (47.2%), respectively. Further, the number of patients who had achieved an MCID in the ODI and EQ-5D score at discharge was 146 (63.2%) and 101 (43.7%), respectively. Moreover, the number of patients who achieved an MCID in the ODI and EQ-5D scores at follow-up was 79 (74.5%) and 45 (42.5%), respectively.

Patients with FBSS with higher NRS scores for LBP at admission had a higher OR for achieving MCID (ORs for achieving MCID 1 and 2 were 1.45 (95% CI: 1.10–1.90) and 3.43 (95% CI: 2.26–5.22), respectively). There was an OR of 2.51 (95% CI: 1.07–5.90) for achieving an MCID in the EQ-5D calculated based on responses provided in the follow-up questionnaire survey regarding the NRS score for LBP at admission. There was an OR of 1.06 (95% CI: 1.03–1.09) for achieving an MCID in the ODI score at admission. There were ORs of 1.16 (95% CI: 1.07–1.27) and 0.85 (95% CI: 0.75–0.96) for achieving an MCID in the ODI and EQ-5D MCID scores, respectively, at the follow-up questionnaire survey. There were ORs of 0.82 (95% CI: 0.77–0.87) and 0.78 (95% CI: 0.69–0.88) for achieving MCID in the EQ-5D score at discharge and the follow-up questionnaire survey, respectively.

Regarding the surgery type, the OR for achieving MCID in the ODI score at discharge in patients who underwent laminectomy was 3.20 (95% CI: 1.02–10.06). The OR for achieving MCID 2 in the NRS score for LBP at discharge in patients who underwent spinal fusion was 3.13 (95% CI: 1.03–9.51). Furthermore, the OR for achieving MCID 1 in the NRS score for LBP at discharge in patients who underwent discectomy was 2.03 (95% CI: 1.07–3.85). Regarding the medical classification based on lumbar MRI or CT findings, the OR for achieving MCID 1 and 2 in the NRS score for LBP at discharge in patients with HNP was 2.29 (95% CI: 1.18–4.47) and 2.97 (95% CI: 1.38–6.39), respectively. There were ORs of 0.29 (95% CI: 0.08–0.99) and 0.04 (95% CI: 0.00–0.37) for achieving an MCID in the ODI and EQ-5D scores, respectively, at discharge in patients with spondylolisthesis. With respect to smoking, the ORs for achieving MCID 1 and 2 in the NRS score at discharge were 0.27 (95% CI: 0.11–0.69) and 0.15 (95% CI: 0.05–0.46), respectively (Table 3).

### 3.5. Treatments

Regarding the inpatient treatment, the patients received the following integrative KM treatments at our hospital: herbal medicine, acupuncture, pharmacopuncture, Chuna manual therapy, and Korean physical therapy (including traction therapy and herbal steam therapy). Integrative KM treatments mainly comprised of those used for lumbar disc herniation and stenosis [27,28]. All the patients underwent acupuncture and pharmacopuncture, which were performed an average of 52.1 ± 32.1 and 51.2 ± 32.3 times, respectively. Herbal medicine was prescribed to 232 (99.1%) patients for an average of 82.0 ± 50.6 times. Chuna manual therapy was prescribed to 220 (94.0%) patients for an average of 26.1 ± 16.9 times (Appendix A).

### 3.6. Follow-Up Survey

The median duration from onset to spinal surgery was 8 (2–52) weeks, with ≥ 2 years (*n* = 23, 21.7%) and < 1 week (*n* = 21, 19.8%) being the most common responses. There were 17 (16.0%) patients who experienced persistent pain immediately after the initial surgery, with the median duration to recurrence being 12 (1–48) months and ≥ 1 year– < 3 years (*n* = 23, 21.7%) being the most common responses. Regarding the reasons for surgery, “no other choice due to extreme pain” (*n* = 65, 61.3%) and “expectation of significant pain reduction” (*n* = 47, 44.3%) were the most common responses. Regarding the reason for satisfaction with spinal surgery, “significant pain reduction” (*n* = 46, 43.4%) and “fast pain reduction” (*n* = 25, 23.6%) were the most common responses. Contrastingly, regarding the reasons for dissatisfaction with spinal surgery, “no pain reduction” (*n* = 44, 41.5%) was the most common response, followed by “sequela” and “frequent pain recurrence” (both *n* = 39, 36.8%). Further, postoperative pain recurrence was observed mostly at the “same level” (62.3%) or “contiguous level” (29.2%). Additionally, 40 (37.7%) patients with FBSS were recommended for reoperation; among them, 16 (40.0%) patients underwent reoperation before being admitted to the KM hospital (Table 4).

Regarding the patients′ global impression of change (PGIC) for KM treatment, 101 (95.3%) patients chose “minimally improved” or better. The mean score for the satisfaction degree with KM treatment was 7.97 ± 1.98 points. Most patients with FBSS were satisfied with pharmacopuncture (*n* = 68, 64.2%), followed by Chuna manual therapy (*n* = 40, 37.7%) and acupuncture (*n* = 34, 32.1%). The most common reasons for satisfaction with KM treatment were “significant pain reduction” (*n* = 37, 34.9%) and “functional recovery” (*n* = 29, 27.4%). The most common reason for dissatisfaction with KM treatment was “cost burden” (*n* = 27, 25.5%). Further, 13 (12.3%) patients underwent reoperation after inpatient KM treatment (Table 5).

## 4. Discussion

This study used various indicators, including pain, functional impairment, HRQoL, and satisfaction degree for comprehensive evaluation of the patients’ status. Most patients admitted to KM hospitals for pain treatment after back surgery had complaints of moderate-to-severe pain at the admission time. KM treatment led to pain reduction, as well as improvement in function and HRQoL. The follow-up questionnaire survey confirmed that these effects showed long-term persistence; moreover, it revealed the reasons for satisfaction/dissatisfaction with surgical and KM treatments.

In 2014, a study on 707 patients with FBSS who received KM treatment [13] reported that most patients (70.4%) were chronic patients with the onset at ≥6 months. Contrastingly, in our study, most patients were acute patients (43.2%) with the onset at <1 month, followed by subacute patients (37.2%). Moreover, chronic patients showed the lowest percentage (19.7%) with the onset at ≥6 months. Therefore, compared with the previous study, the present study observed a slight decrease in the duration from postoperative pain recurrence to choosing KM treatment among patients with FBSS. Further, the length of hospital stay increased by approximately 1 week in our study (28.1 ± 17.1 days) compared with the previous study (21.72 ± 11.91 days). The most common surgery types and operated disc level were laminectomy (93.6%) and L4–5 (67.1%), which was consistent with previous findings on KM treatment for patients with FBSS [13,14].

There was a significant decrease in the NRS scores for LBP and leg pain at admission and discharge (*p* < 0.001) with LBP showing a greater improvement than the MCID. Further, 41.6% of the patients showed ≥50% decrease in the NRS score, which was slightly lower than that reported by previous studies [14,26]. This inconsistency could be attributed to the fact that previous studies assessed 6-month treatment while the present study had a relatively shorter treatment period of 4 weeks. There was a significant post-treatment improvement in the ODI (functional impairment) and EQ-5D scores (HRQoL), with the follow-up questionnaire survey confirming that these were long-term effects. This is consistent with previous findings of a decrease in the ODI (11 [29], 16 [23], 19.4 [30]) after spinal cord stimulation treatment in patients with FBSS; however, we observed much greater changes in the EQ-5D scores (0.18 [29], 0.174 [23], 0.16 [30]).

Analysis of data obtained at admission and discharge from patients with FBSS revealed that the mean changes in the NRS scores for LBP, ODI, and EQ-5D scores, but not the NRS score for leg pain, were higher than the MCID. Analysis regarding MCID achievement revealed that patients with relatively severe FBSS who had high ODI and NRS scores for LBP or low EQ-5D scores who received integrative KM treatment showed a higher OR for achieving the MCID in each indicator. Patients with higher NRS scores for LBP at admission showed greater MCID achievement in the HRQoL at follow-up. Patients with relatively severe pain are thought to achieve greater improvement, which results in better results for current HRQoL.

Regarding lumbar MRI or CT findings, patients with FBSS with HNP showed a higher OR of achieving MCID in the NRS score for LBP. This is because integrative KM treatment in patients with lumbar disc herniation has a long-term effect on disc resorption [31], as well as improvement in pain and function [32,33]. The principle underlying the improvement of pain caused by lumbar disc herniation is suggested by previous studies on the effects of Shinbaro [34,35]. GCSB-5, which is the main component of Shinbaro, exerts anti-inflammatory effects [36], neuroprotection, and nerve regeneration effects [37]. Additionally, such effects influence tissue regeneration, as well as reduction of inflammation and pain induced by lumbar disc herniation.

Contrastingly, patients with FBSS with spondylolisthesis showed a low OR for achieving MCID in terms of functional impairment and HRQoL. Generally, conservative treatment for spondylolisthesis in adults with progressing degeneration is not as effective as surgical treatment [38,39], which could negatively affect changes in functional impairment and HRQoL in these patients.

Items in the follow-up questionnaire survey included information regarding surgical treatment, satisfaction/dissatisfaction with surgical treatment, and satisfaction/dissatisfaction with KM treatment, which could not be easily identified from the data collected from the EMR. The questionnaire survey indicated that the most common reason for choosing surgical treatment was “significant pain reduction”, which was the same reason for satisfaction. Surgical treatment reduced acute pain; however, there was a high likelihood of residual pain or pain recurrence. Consequently, the patients were dissatisfied with their physical condition after surgical treatment. However, patients with previous failure using generic conservative treatments, including drug therapy or physical therapy, recognized that only a few treatment options remained after surgical treatment. Therefore, they considered reoperation to resolve their symptoms. Integrative KM treatment provides an alternative for broadening the treatment options for such patients with FBSS.

Although PGIC for KM treatment was measured after a long post-discharge period, most responses were positive with no response indicating a negative change. The reasons for satisfaction with KM treatment included significant pain reduction and functional recovery. Among the KM treatment modalities, pharmacopuncture showed the highest satisfaction degree (64.2%), followed by Chuna manual therapy (37.7%), and acupuncture (32.1%), which were the most commonly administered treatment modalities within three months after the questionnaire survey. These findings indicated that most patients receiving inpatient KM treatment and answered to the follow-up survey had a high satisfaction degree with and positive impression of KM treatment. However, the response rate to the survey was less than 50%, the interpretation need to be done carefully.

Pharmacopuncture, which showed the highest satisfaction degree, involves a combination of conventional acupuncture and herbal medicine to sustain their mechanical and chemical effects through optimal acupoint access [40]. Currently, pharmacopuncture is applied for various diseases and has shown excellent efficacy to treat musculoskeletal disorders and obesity [41]. Moreover, it has been used to alleviate symptoms in patients with cancer and stroke [42,43], as well as to treat peripheral neuropathy [44] and asthma [45]. Additionally, several studies have confirmed its safety [46,47]. Chuna manual therapy, which showed the highest degree of satisfaction after pharmacopuncture, is applied to recover structural balance and lumbar function. Chuna manual therapy is based on the KM principle of a mutual association of physical structure with function and is a manual therapy for orthopedic structural balance and functional recovery [48]. Integrative KM treatment of patients with FBSS showed pain reduction and significant functional improvement with Chuna manual therapy being considered to contribute to the structural and functional recovery. Generally, acupuncture is used to manage acute and chronic pain [49,50]. Despite some methodological limitations, acupuncture is known to be clinically effective in reducing pain after back surgery [51].

This study combined a retrospective analysis of EMR and long-term follow-up questionnaire survey. Given the limitations resulting from this study design, the study could not be conducted under a controlled environment, and there was no control group for comparison. In the questionnaire survey, the question regarding the intraoperative status relied on the patient’s long-term memory; therefore, recall inaccuracy cannot be dismissed. Moreover, we included patients with a history of at least one back surgery; therefore, there was some uncertainty regarding distinction between pain experience and tension/pain resulting from overuse, as well as other chronic LBP disorders, including myofascial pain syndrome, fibromyalgia, and CRPS. In this study, the response rate to the follow-up survey was less than 50%, so we need to interpret the efficacy of integrative KM treatment with caution. Finally, the applied integrative KM treatment combines multiple treatment modalities; therefore, it is difficult to differentiate among the individual effects of each treatment modality.

Despite these limitations, this study is significant since it analyzed the long-term effects of integrative KM treatment for FBSS thorough a follow-up survey. A previous study explored the same research topic using 707 patients with FBSS [13]; however, it presented statistics regarding the use of integrative KM treatment in patients with FBSS and did not determine the therapeutic effects on pain, functional impairment, and HRQoL. Contrastingly, this study accounted for the realistic clinical effects of integrative KM treatment by comparing the pre- and post-treatment pain level, as well as the conditions of patients with FBSS who received integrative KM treatment. Additionally, it determined the long-term therapeutic effects and degree of satisfaction with treatment. A prospective 1-year observational study on 120 patients [14] was conducted under relatively controlled settings; however, it only included outpatients. Contrastingly, this study only included patients who were hospitalized for at least one week; therefore, we could have included patients with more severe conditions, which allowed comprehensive assessment of the prognosis of patients with FBSS who received integrative KM treatment through a long-term follow-up survey.

Our findings are favorable since they demonstrate the therapeutic effect and satisfaction degree with integrative KM treatment in patients with FBSS in South Korea. Therefore, integrative KM treatment could be an alternative treatment option for patients with FBSS, who currently have limited treatment options. The multi-dimensional effectiveness of integrative KM treatment in patients with FBSS was tested using proven tools for pain, functional impairment, and HRQoL, with the findings demonstrating long-term maintenance of the treatment effects. However, there is a need for more evidence regarding integrative KM treatment for patients with FBSS and additional validation through well-designed RCTs with appropriate samples.

## 5. Conclusions

Integrative KM treatment could reduce persistent or recurrent pain, as well as improve functional impairment and HRQoL in patients with FBSS. Therefore, integrative KM treatment could be considered an effective conservative treatment modality for managing patients with FBSS.

## Figures and Tables

**Figure 1 jcm-10-01703-f001:**
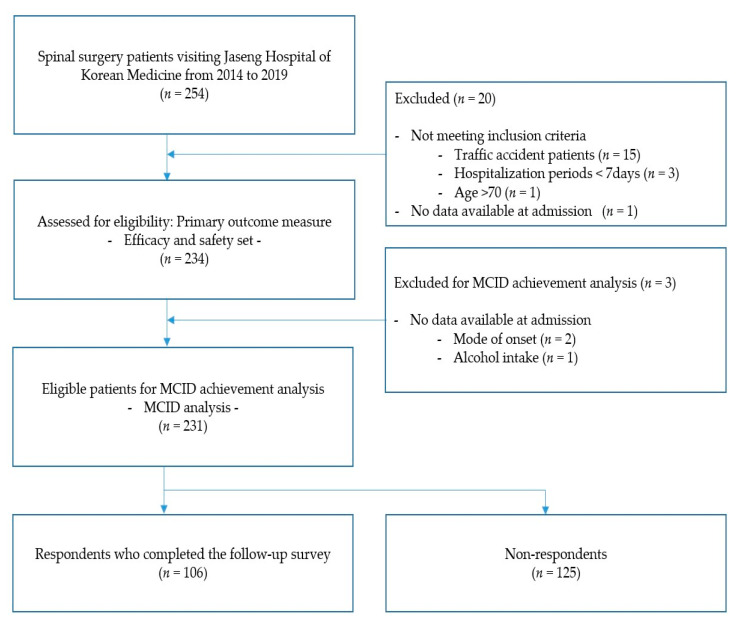
Study flow of participants. MCID: minimal clinically important difference.

**Figure 2 jcm-10-01703-f002:**
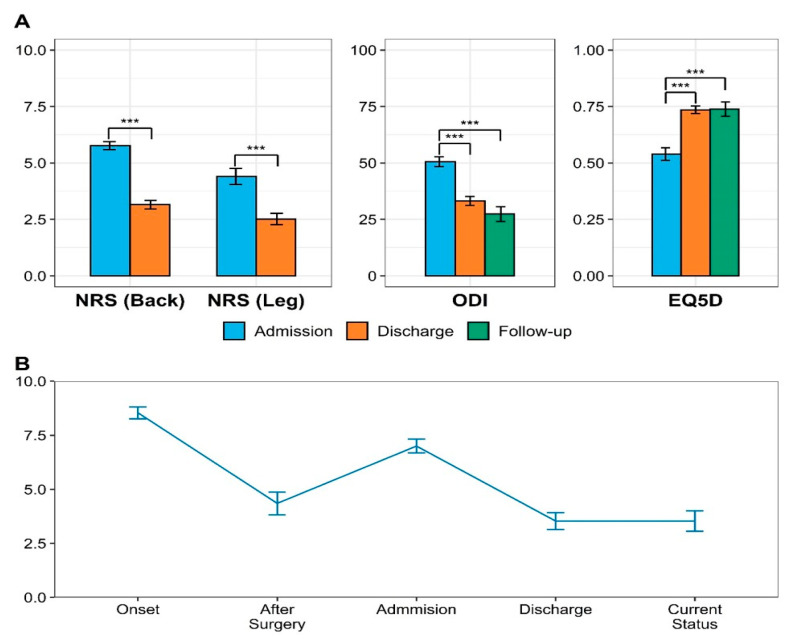
Post-treatment changes of main outcomes. (**A**) NRS, ODI, and EQ-5D scores of patients with FBSS. *** *p*-Value < 0.001. (**B**) the mean NRS score at onset, after surgery, before admission to KM hospital, after discharge from KM hospital in the follow-up questionnaire survey. NRS: numeric rating scale; ODI: Oswestry disability index; EQ-5D: EuroQol 5-dimension.

**Table 1 jcm-10-01703-t001:** Baseline Characteristics of Patients with FBSS (*n* = 234).

	Mean ± SD or Median (%)
**Age**	
Mean ± SD	54.9 ± 11.5
20≤, <30	3 (1.3)
30≤, <40	27 (11.5)
40≤, <50	49 (20.9)
50≤, <60	54 (23.1)
60≤, <70	89 (38.0)
70≤	12 (5.1)
**Sex**	
Male	93 (39.7)
Female	141 (60.3)
**Length of hospital stay**	
Mean ± SD (day)	28.1 ± 17.1
Median (IQR)	23.5 (14.3–38)
**Type of spinal surgery ^1^**	
Laminectomy	219 (93.6)
Discectomy	71 (30.3)
Spinal fusion	31 (13.2)
Vertebroplasty	12 (5.1)
Artificial disc replacement	2 (0.9)
**Operated disc levels** **^1^**	
L1–2	9 (3.8)
L2–3	13 (5.6)
L3–4	36 (15.9)
L4–5	157 (67.1)
L5-S1	84 (35.9)
**Duration from pain recurrence until admission**	
Mean ± SD (day)	205.4 ± 484.2
Median (IQR)	39 (7–147.3)
**Mode of pain recurrence**	
No specific cause	150 (64.1)
Overwork/Over exercise	42 (17.9)
Trauma/Fall	33 (14.1)
Other	7 (3.0)
Unknown	2 (0.9)
**Analgesics ^†^**	
Mean ± SD	3.6 ± 10.5
Median (IQR)	0 (0,1,2,3)
**Steroid injections ^†^**	
Mean ± SD	0.9 ± 3.5
Median (IQR)	0 (0,1)
**Radiological findings of MRI/CT scans ^1^**	
Herniation of the nucleus pulposus	129 (55.1)
Protrusion	77 (32.9)
Extrusion	77 (32.9)
Sequestration	0 (0.0)
Spinal stenosis	67 (28.6)
Central canal	42 (17.9)
Foraminal	42 (17.9)
Spondylolisthesis	17 (7.3)
Vertebral fracture	11 (4.7)
Other	2 (0.9)
**Comorbidity ^1^**	
Hypertension	37 (15.8)
Diabetes mellitus	22 (9.4)
Cardiovascular disease	44 (18.8)
Thyroid-related comorbidity	10 (4.3)
Liver-related comorbidity	4 (1.7)
Other	63 (26.9)
**Smoking**	
Yes	44 (18.8)
No	190 (81.2)
**Alcohol intake**	
Yes	44 (18.8)
No	189 (80.8)
Unknown	1 (0.4)
**Occupation**	
Unemployed ^‡^	133 (56.8)
Office work ^§^	63 (26.9)
Service or retail industry ^11^	22 (9.4)
Manual labor **^††^**	15 (6.4)
Unknown	1 (0.4)

^1^ Multiple check. ^†^ Treatments before admission. ^‡^ Housewife/student/retired. ^§^ Office worker/manager/public servant/professional practitioner. 11 Self-employed/service or retail industry worker. ^††^ Agriculture, forestry, fishery, or mining industry worker/equipment mechanic or machinery operator/professional soldier. FBSS: failed back surgery syndrome; SD: standard deviation; IQR: interquartile range; MRI: magnetic resonance imaging; CT: computed tomography.

**Table 2 jcm-10-01703-t002:** Post-treatment changes in the values.

Outcome		Outcome	Value	Difference	*p*-Value
NRS	LBP	Admission	5.77 ± 1.38	—	
		Discharge	3.15 ± 1.49	−2.62 (−2.82, −2.41)	<0.001
	Leg pain	Admission	4.40 ± 2.76	—	
		Discharge	2.52 ± 1.99	−1.88 (−2.08, −1.68)	<0.001
ODI		Admission	50.55 ± 16.53	—	
		Discharge	33.19 ± 15.93	−17.35 (−19.62, −15.09)	<0.001
		Follow-up	27.39 ± 16.97	−23.50 (−26.39, −20.61)	<0.001
EQ-5D		Admission	0.54 ± 0.22	—	
		Discharge	0.74 ± 0.13	0.20 (0.17, 0.22)	<0.001
		Follow-up	0.74 ± 0.17	0.20 (0.17, 0.24)	<0.001

Values are presented as mean ± SD. Differences are expressed as mean change (95% CI) compared with the admission baseline value. NRS: numeric rating scale; LBP: low back pain; ODI: Oswestry disability index; EQ-5D: EuroQol 5-dimension; SD: standard deviation; CI: confidence interval.

**Table 3 jcm-10-01703-t003:** Analysis of MCID achieved by patients with FBSS.

		Discharge (*n* = 231)				Follow-Up (*n* = 106)		
		MCID 1 of LBP NRS	MCID 2 of LBP NRS	MCID of ODI	MCID of EQ-5D	MCID of ODI	MCID of EQ-5D	Reoperation
Case (%)		96 (41.6)	109 (47.2)	146 (63.2)	101 (43.7)	79 (74.5)	45 (42.5)	13 (12.3)
Age	20≤, <40	—	—	—	—	—	—	—
	40≤, <60	0.60 (0.23, 1.53)	0.61 (0.20, 1.84)	0.69 (0.24, 2.03)	2.38 (0.43, 13.2)	3.64 (0.57, 23.23)	3.11 (0.20, 49.16)	0.30 (0.03, 3.07)
	60≤	0.67 (0.24, 1.91)	0.86 (0.25, 2.96)	0.62 (0.19, 2.01)	4.61 (0.74, 28.88)	2.17 (0.23, 20.54)	6.50 (0.26, 160.73)	0.66 (0.06, 7.45)
Sex	Female	0.72 (0.34, 1.49)	0.6 (0.25, 1.44)	0.79 (0.35, 1.79)	1.21 (0.40, 3.66)	0.48 (0.07, 3.42)	0.39 (0.04, 3.59)	1.76 (0.16, 19.39)
Onset	<1 month	—	—	—	—	—	—	—
	1≤, <6 months	1.36 (0.68, 2.72)	1.04 (0.48, 2.24)	1.05 (0.51, 2.17)	1.38 (0.44, 4.34)	0.19 (0.03, 1.05)	0.10 (0.01, 0.82)	8.41 (1.05, 67.57)
	6 months <	0.89 (0.37, 2.13)	0.76 (0.29, 2.00)	1.02 (0.41, 2.49)	0.84 (0.22, 3.28)	0.22 (0.03, 1.76)	0.07 (0.01, 0.75)	3.99 (0.32, 50.13)
Mode of Onset	Overwork/Over exercise	1.65 (0.74, 3.68)	1.33 (0.52, 3.38)	0.62 (0.26, 1.47)	1.92 (0.51, 7.13)	1.52 (0.19, 12.03)	2.18 (0.21, 22.65)	1.09 (0.07, 16.21)
	Trauma/Fall	2.00 (0.85, 4.71)	2.01 (0.74, 5.42)	0.99 (0.39, 2.49)	1.09 (0.25, 4.69)	1.40 (0.20, 9.65)	0.55 (0.05, 5.59)	6.42 (0.92, 44.9)
Surgery type	Laminectomy	1.48 (0.48, 4.57)	1.49 (0.42, 5.33)	3.20 (1.02, 10.06)	7.71 (1.13, 52.37)	0.48 (0.05, 4.84)	0.35 (0.02, 5.33)	1.09 (0.08, 15.53)
	Spinal fusion	2.01 (0.76, 5.28)	3.13 (1.03, 9.51)	0.59 (0.23, 1.54)	0.31 (0.07, 1.40)	0.18 (0.03, 1.12)	0.37 (0.02, 8.20)	2.61 (0.38, 17.81)
	Discectomy	2.03 (1.07, 3.85)	1.73 (0.84, 3.56)	0.75 (0.36, 1.54)	1.30 (0.46, 3.72)	1.54 (0.30, 8.02)	2.58 (0.38, 17.68)	1.05 (0.17, 6.35)
Operated disc level	L4-5	0.85 (0.43, 1.67)	1.30 (0.61, 2.77)	0.95 (0.46, 1.96)	0.70 (0.25, 1.95)	1.02 (0.23, 4.60)	0.11 (0.01, 0.95)	4.96 (0.66, 37.14)
Radiological findings	HNP	2.29 (1.18, 4.47)	2.97 (1.38, 6.39)	1.47 (0.73, 2.94)	1.34 (0.47, 3.82)	1.80 (0.36, 8.96)	1.22 (0.21, 6.98)	0.26 (0.04, 1.85)
	Stenosis central canal	1.41 (0.71, 2.82)	1.78 (0.81, 3.90)	1.07 (0.52, 2.20)	0.41 (0.14, 1.23)	2.82 (0.57, 13.82)	2.03 (0.20, 21.10)	0.80 (0.14, 4.58)
	Spondylolisthesis	0.63 (0.18, 2.18)	0.27 (0.06, 1.16)	0.29 (0.08, 0.99)	0.04 (0, 0.37)	1.33 (0.11, 15.51)	9.51 (0.33, 278.24)	3.73 (0.29, 48.21)
Comorbidity	Hypertension	0.59 (0.25, 1.40)	0.73 (0.27, 2.00)	0.70 (0.29, 1.67)	0.45 (0.12, 1.72)	0.45 (0.08, 2.59)	0.75 (0.08, 6.99)	3.91 (0.44, 35.12)
	Diabetes mellitus	1.19 (0.42, 3.41)	0.46 (0.13, 1.67)	0.38 (0.13, 1.17)	0.12 (0.02, 0.72)	0.13 (0.01, 1.41)	0.34 (0.01, 17.81)	0.15 (0.01, 2.63)
Social history	Smoking	0.27 (0.11, 0.69)	0.15 (0.05, 0.46)	0.52 (0.19, 1.40)	0.47 (0.11, 2.02)	1.43 (0.12, 16.53)	1.82 (0.10, 33.60)	2.02 (0.13, 30.51)
	Alcohol intake	1.08 (0.47, 2.47)	1.46 (0.55, 3.84)	1.31 (0.55, 3.13)	1.60 (0.45, 5.74)	3.64 (0.40, 33.28)	5.27 (0.49, 57.31)	1.01 (0.10, 10.13)
NRS at admission	LBP	1.45 (1.10, 1.90)	3.43 (2.26, 5.22)	1.05 (0.80, 1.37)	0.80 (0.53, 1.20)	1.06 (0.55, 2.08)	2.51 (1.07, 5.90)	0.63 (0.32, 1.28)
	Leg pain	0.97 (0.87, 1.09)	0.88 (0.76, 1.01)	0.93 (0.81, 1.05)	0.86 (0.70, 1.05)	0.87 (0.64, 1.18)	0.84 (0.59, 1.18)	1.25 (0.85, 1.83)
ODI at admission		1.01 (0.98, 1.03)	1.00 (0.97, 1.03)	1.06 (1.03, 1.09)	0.99 (0.95, 1.04)	1.16 (1.07, 1.27)	0.85 (0.75, 0.96)	0.97 (0.90, 1.05)
EQ-5D at admission		1.00 (0.98, 1.02)	1.00 (0.98, 1.02)	0.99 (0.97, 1.01)	0.82 (0.77, 0.87)	1.03 (0.98, 1.08)	0.78 (0.69, 0.88)	0.99 (0.94, 1.04)
AUC		0.73 (0.66, 0.80)	0.84 (0.79, 0.89)	0.78 (0.72, 0.84)	0.95 (0.93, 0.98)	0.89 (0.81, 0.97)	0.97 (0.94, 1.00)	0.84 (0.76, 0.93)

The results are presented as odds ratio and 95% CI. Each outcome is defined as MCID achievement. The MCID criteria are as follows: a reduction in the NRS score of LBP by ≥50% or NRS score of 2.25, ODI score of 9, and EQ-5D score of 0.17. FBSS: failed back surgery syndrome; MCID: minimal clinically important difference; LBP: low back pain; NRS: numeric rating scale; ODI: Oswestry disability index; EQ-5D: EuroQol 5-dimension; HNP: herniation of the nucleus pulposus; AUC: area under the curve; CI: confidence interval.

**Table 4 jcm-10-01703-t004:** Follow-up survey of spinal surgery (*n* = 106).

	Mean ± SD or Median (%)
**Duration from the onset to spinal surgery**	
Mean ± SD (weeks)	57.9 ± 109.5
Median (IQR)	8 (2–52)
≤1 week	21 (19.8)
<1 week, <1 month	12 (11.3)
≤1 month, <2 months	17 (16.0)
≤2 months, <6 months	17 (16.0)
≤6 months, <1 year	3 (2.8)
≤1 year, <2 years	11 (10.4)
≤2 years	23 (21.7)
Unknown	2 (1.9)
**Duration from the first spinal surgery until pain recurrence**	
Mean ± SD (month)	40.0 ± 69.1
Median (IQR)	12 (1–48)
Continuation	17 (16.0)
≤1 month	16 (15.1)
<1 month, <1 year	11 (10.4)
≤1 year, <3 years	23 (21.7)
≤3 years, <5 years	15 (14.2)
≤5 years, <10 years	12 (11.3)
≤10 years	12 (11.3)
**Duration from discharge until follow-up survey**	
Mean ± SD (months)	34.54 ± 14.47
Median (IQR)	34.95 (22.35–46.85)
**Number of spinal surgeries**	1.3 ± 0.5
**Reasons for spinal surgery ^1^**	
No resistance due to significant pain	65 (61.3)
Expectation of a significant pain reduction	47 (44.3)
Compelled by the doctor	25 (23.6)
Few sequelae	10 (9.4)
No information regarding other treatments	9 (8.5)
Low cost burden	7 (6.6)
Easy rehabilitation	6 (5.7)
Recommended by the people around	2 (1.9)
Other	3 (2.8)
**Satisfactory reason with spinal surgery ^1^**	
Significant pain reduction	46 (43.4)
Fast pain reduction	25 (23.6)
A short treatment period	12 (11.3)
Low cost burden	12 (11.3)
Few sequelae	11 (10.4)
Sufficient information regarding the surgery	6 (5.7)
Easy rehabilitation	3 (2.8)
Other	2 (1.9)
**Unsatisfactory reason for spinal surgery ^1^**	
No pain reduction	44 (41.5)
Sequela	39 (36.8)
Frequent pain recurrence	39 (36.8)
Anxiety	9 (8.5)
No functional recovery	8 (7.5)
Painful treatment	5 (4.7)
Cost burden	5 (4.7)
Insufficient information regarding the surgery	5 (4.7)
Other	1 (0.9)
**Recurrence at operated disc level**	
Same level	66 (62.3)
Contiguous level	31 (29.2)
Different level	5 (4.7)
Unknown	4 (3.8)
**Recommendation for reoperation ^†^**	
Yes	40 (37.7)
No	66 (62.3)
**Reoperation ^†^**	
Yes	16 (40.0)
No	24 (60.0)

^1^ Multiple answers allowed. ^†^ Reoperation before admission to the Korean medicine hospital. SD: standard deviation; IQR: interquartile range.

**Table 5 jcm-10-01703-t005:** Follow-up survey of Korean medicine treatment (*n* = 106).

	Mean ± SD or Median (%)
**PGIC for Korean medicine treatment**	
Very much improved	21 (19.8)
Much improved	43 (40.6)
Minimally improved	37 (34.9)
No change	5 (4.7)
Minimally worse	0 (0)
Much worse	0 (0)
Very much worse	0 (0)
**Satisfaction degree of Korean medicine treatment ^1^**	7.97 ± 1.98
**Korean medicine treatment with highest satisfaction degree ^†^**	
Pharmacopuncture	68 (64.2)
Chuna manual therapy	40 (37.7)
Acupuncture	34 (32.1)
Herbal medicine	28 (26.4)
Korean physical therapy	20 (18.9)
Cupping	7 (6.6)
Other	17 (16.0)
**Satisfactory reason with Korean medicine ^†^**	
Significant pain reduction	37 (34.9)
Functional recovery	29 (27.4)
Fast pain reduction	27 (25.5)
Not painful treatment	27 (25.5)
Sufficient information regarding treatments	27 (25.5)
Low recurrence	14 (13.2)
Diverse treatment	11 (10.4)
Low cost burden	0 (0)
Other	7 (6.5)
**Unsatisfactory reason with Korean medicine ^†^**	
Cost burden	27 (25.5)
Prolonged treatment period	16 (15.1)
Frequent pain recurrence	8 (7.5)
No pain reduction	6 (5.7)
Insufficient information regarding treatments	1 (9)
Unnecessarily high number of treatment types	0 (0)
Painful treatment	0 (0)
Other	6 (5.7)
**Reoperation after taking Korean medicine treatment**	
Yes	13 (12.3)
No	93 (87.7)
**Number of reoperations after taking Korean medicine treatment**	1.5 ± 1.4
**Present treatment within 3 months**	
Yes	45 (42.5)
No	61 (57.5)
**Type of present treatment ^†^**	
Acupuncture	19 (17.9)
Pharmacopuncture	15 (14.2)
Chuna manual therapy	11 (10.4)
Herbal medicine	10 (9.4)
Cupping	6 (5.7)
Manual therapy	13 (12.3)
Physical therapy (ESWT, TENS, ICT)	10 (9.4)
Steroid injection	8 (7.5)
Medication	8 (7.5)
Therapeutic exercise	5 (4.7)
Other	1 (0.9)

^1^ 1 ≤, ≤ 10. ^†^ Multiple answers allowed. PGIC: patients′ global impression of change; SD: standard deviation; ESWT: extracorporeal shock wave therapy; TENS: transcutaneous electrical nerve stimulation; ICT: interferential current therapy.

## Data Availability

The data presented in this study are available upon request from the corresponding author. The data are not publicly available due to privacy/ethical restrictions.

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
