# Peer review of "Long-Term Follow-Up of Inpatients with Failed Back Surgery Syndrome Who Received Integrative Korean Medicine Treatment: A Retrospective Analysis and Questionnaire Survey Study"

_jcm, 2021, doi:10.3390/jcm10081703_

Round 1

Reviewer 1 Report

Overall an interesting study, which would benefit from a bit more methodological detail and a more considered discussion. There are some fundamental flaws in the methodology that make it difficult to make definitive conclusions. Be careful to ensure that the conclusions you draw are supported by the science and not by your personal beliefs

Lines 39 – 42 : these numbers are misleading as they represent all spinal fusions, Not just lumbar ones. As you are focusing your paper on low back pain then the spinal surgery figures should also focus on this condition.

Line 94 – 95  “Patients with a history of low back surgery admitted with a chief complaint of persistent pain/discomfort or recurrent pain/discomfort after alleviation” please edit as it doesn’t make sense – what does “after alleviation” mean?? I assume that there was some means to ensure that the pain they present with was related to the same area as the surgery – if so this needs to be made clear.

Lines 113 – 114 “ 7) Patients unfit for study participation due to other reasons as determined by the  researcher” this seems unnecessarily vague – can you provide examples?

Lines 224 – 225 “We selected 254 patients who experienced recurrent LBP or leg pain after back surgery with subsequent admission to one of four regional network KM hospitals” – How were they selected for recruitment into the study – was this group selected at random, via consecutive recruitment or selectively recruited??

I am slightly confused with the recruitment so may need to be clarified. These are patients who had spinal surgery in the past, who present with spinal pain (I assume in the same area) – which you are classifying as FBSS – so all patients in this study had FBSS  - they went to hospital for an average of 28.1 days. My confusion I think sits with the line 335 – 336 – I assume that this information i.e. amount of time from surgery to onset (of symptoms I assume) would have been collected from the case notes or EMRs – why would you collect this from a follow up survey. How long after discharge were the surveys done?? Were the outcome measures collected via interview in the EMRs or via papercopy?? If they were different do you feel this may have affected the validity of comparing the results of EMR to interview?

Given the significant drop out rate on the follow up it is dangerous to make definitive statements on the benefits of KM – given you only have data from 45% of patients – It is usual for this type of study to do Intention to Treat analysis to explore effectiveness. Given less then 50% of patients responded you cannot make the clam that “… most patients receiving inpatient KM treatment had a high satisfaction degree with and positive impression of KM treatment” .

Lines 417 – 418 : I am unsure how “protective effects on joint cartilage against osteoarthritis progression” could be used to describe how a change in 28 days in pain from a  lumbar disc herniation occurred.

Line 463 – 464 “This study combined a retrospective analysis of EMR and long-term follow-up questionnaire survey” How long past the inpatient visit was the survey collected??

Line 474 : “Despite these limitations, this study is significant since it analysed the long-term effects of integrative KM treatment for FBSS thorough a follow-up survey.” – Less than 50 % of patients responded (for all we know the 55% of patients who did not respond were significantly worse off on the long term with KM – you have no evidence to refute this so be careful with the statements – you have no control group and all patients knew what they were getting (i.e. they were not blinded) and so you cannot rule out placebo and maturation bias – I am not saying the study results are not interesting but be very careful about making definitive statements not supported by the evidence

Reviewer 2 Report

The author(s) of “Long-term follow-up of inpatients with failed back surgery syndrome who received integrative Korean medicine treatment: A retrospective analysis and questionnaire survey study” broached the subject, which is a largely uncharted territory. The targeted receivers of that paper are not certainly limited to neurosurgeons and pain therapists. I believe that many of future studies would cite this paper. The narrative introduction evenly leads a reader to the core of the manuscript. Well-written sections are additionally enriched by the detailed tables and figure. The authors proved that KM could stand as a salvage treatment for pts with potentially failed response to conventional treatment of FBSS. The results bring a novelty to existing knowledge.

There are some minor issues to be solved before final publishing:

  1. Ethical concerns: appreciating the published researches originating at Jaseng Hospital of Korean Medicine, it would be plausible to provide more details on IRB. In current study IRB approval has been obtained by the researchers (No. JASENG 2020-09-017). However, I could not find the website of the IRB.
  2. Low rate of patients who responded to the survey should be mentioned in the Discussion section.

Based on the above suggestions the author(s) should correct and resubmit the revised manuscript.

I’m willing to review the improved version of this manuscript. Thank you.
